# Ultrasound-Guided Prolotherapy for Sciatica Secondary to Sacrospinous Ligament Calcification: A Potential and Previously Overlooked Etiological Factor in Deep Gluteal Syndrome—A Case Report and Literature Review

**DOI:** 10.3390/life15091486

**Published:** 2025-09-22

**Authors:** Yonghyun Yoon, King Hei Stanley Lam, Jaeyoung Lee, Rowook Park, Jaehyun Shim, Jonghyeok Lee, Daniel Chiung-Jui Su, Kenneth Dean Reeves, Stephen Cavallino

**Affiliations:** 1Department of Orthopaedic Surgery, Gangnam Sacred Heart Hospital, Hallym University College of Medicine, 1 Singil-ro, Yeongdeungpo-gu, Seoul 07441, Republic of Korea; 2Incheon Terminal Orthopedic Surgery Clinic, Inha-ro 489beon-gil, Namdong-gu, Incheon 21574, Republic of Korea; 2wo02wo0@naver.com (J.L.); prwook@naver.com (R.P.); perfectceive@gmail.com (J.L.); 3International Academy of Regenerative Medicine, Inha-ro 489beon-gil, Namdong-gu, Incheon 21574, Republic of Korea; 4The Board of Clinical Research, The International Association of Musculoskeletal Medicine, Kowloon, Hong Kong; 5Musculoskeletal Ultrasound (MSKUS), 1035 E. Vista Way #128, Vista, CA 92084, USA; 6Faculty of Medicine, The University of Hong Kong, Hong Kong; 7Faculty of Medicine, The Chinese University of Hong Kong, Hong Kong; 8The Board of Clinical Research, The Hong Kong Institute of Musculoskeletal Medicine, Kowloon, Hong Kong; 9Department of Rehabilitation Medicine, Sae Yonsei Rehabilitation Clinic, Seoul 03186, Republic of Korea; 10Department of Neurosurgery, Chungdammadi Neurosurgery Clinic, Seoul 03186, Republic of Korea; 11Bareun Neurosurgery Clinic, 39, Daenong-ro, Heungdeok-gu, Cheongju-si 28402, Republic of Korea; 12Department of Physical Medicine and Rehabilitation, Chi Mei Medical Center, Tainan 710, Taiwan; dr.daniel@gmail.com; 13Tempo Regeneration Center for Musicians, Tainan 700, Taiwan; 14Rehabilitation Medicine, Private Practice, Kansas City, MO 64132, USA; deanreevesmd@gmail.com; 15European School of Prolotherapy (ESP), 1st Mednikarska str, 1510 Sofia, Bulgaria; s.cavallino@gmail.com; 16Hackett Hemwall Patterson Foundation(HHPF), 7880 Sweeny Rd., Barneveld, WI 53507, USA

**Keywords:** deep gluteal syndrome, sciatica, sacrospinous ligament, ligament calcification, prolotherapy, ultrasound-guided injection, entrapment neuropathy, non-discogenic pain

## Abstract

Background: Deep gluteal syndrome (DGS) is an underdiagnosed cause of sciatica-like pain, involving the entrapment of the sciatic nerve by various structures within the subgluteal space. While cases of ossification or calcification in the context of severe pelvic imbalance have been rarely reported, isolated SSL calcification as a primary cause of DGS remains largely unexplored and undocumented. This case report presents the first documented instance of sacrospinous ligament (SSL) calcification identified as the primary cause of DGS and its successful management with ultrasound-guided prolotherapy. Case Presentation: A 51-year-old female presented with severe, worsening left-sided sciatica of several months’ duration. Physical examination revealed an antalgic gait, positive sacroiliac joint tests, and multiple positive DGS-specific provocative tests (FAIR, Pace sign, Seated Piriformis Stretch). Radiographs and musculoskeletal ultrasound (MSK-US) confirmed calcification within the left sacrospinous ligament, with associated sciatic nerve swelling. The patient underwent three sessions of ultrasound-guided prolotherapy (dextrose 10% with lidocaine) targeting the calcification site, followed by a structured rehabilitation program. Results: The patient reported a significant reduction in pain, from a Visual Analog Scale (VAS) score of 10/10 to 1/10 within one month. All previously positive provocative tests converted to negative, indicating a resolution of the nerve entrapment. Functional mobility was fully restored. Conclusions: This case highlights isolated sacrospinous ligament calcification as a potential and previously overlooked pathological entity responsible for deep gluteal syndrome. To our knowledge, this is the first report to implicate ligamentous calcification as a primary etiological factor in DGS. Musculoskeletal ultrasound proved indispensable for both diagnosis and treatment guidance. Furthermore, ultrasound-guided prolotherapy emerged as a successful and minimally invasive therapeutic option in this case, potentially by stabilizing the ligament and reducing neurogenic inflammation. This case expands the differential diagnosis of sciatica, introduces a new target for intervention in refractory cases, and underscores the need for future studies in larger patient cohorts to validate these findings.

## 1. Introduction

Sciatica, characterized by pain radiating along the sciatic nerve pathway, is a predominant reason for seeking medical care, frequently associated with debilitating low back pain (LBP) [1]. The pain typically follows a dermatomal pattern, most commonly affecting the L5 and S1 nerve root distributions, and can be accompanied by sensory disturbances, motor weakness, and diminished reflexes [2]. The conventional diagnostic paradigm, heavily influenced by imaging findings, predominantly attributes sciatica to compressive radiculopathy stemming from lumbar disk herniation or spinal stenosis [3,4]. This focus often leads to a diagnostic and therapeutic algorithm centered on the lumbar spine, with treatments ranging from conservative management with physical therapy and epidural steroid injections to more invasive surgical interventions like discectomy or laminectomy [5].

However, a substantial and often frustrating proportion of patients, estimated to be between 10% and 40%, experience persistent or recurrent pain despite the absence of significant spinal pathology on advanced imaging or following technically successful surgical intervention for a confirmed disk lesion [6,7]. This clinical conundrum underscores the critical necessity of investigating extra-spinal etiologies for sciatic pain. Failure to identify the true source of pain can lead to a cascade of unnecessary treatments, iatrogenic complications, chronic opioid use, and significant psychological distress, including anxiety and depression, profoundly reducing the patient’s quality of life [8].

The deep gluteal space, a complex and often overlooked anatomical corridor, has been increasingly recognized as a potential site for non-discogenic sciatic nerve entrapment, a condition now collectively termed Deep Gluteal Syndrome (DGS) [9,10]. DGS encompasses a spectrum of disorders where the sciatic nerve is compressed or irritated by various structures within the subgluteal space. These structures include, but are not limited to, a hypertrophied or fibrotic piriformis muscle, congenital or acquired fibrovascular bands, vascular anomalies such as varicosities or persistent sciatic arteries, adhesions from previous surgery or trauma, and other deep gluteal muscles like the gemelli or obturator internus [11,12]. The clinical presentation of DGS often meticulously mimics classic lumbar radiculopathy, featuring buttock pain, posterior thigh pain, and radicular sensations. However, it is distinguished by its failure to respond to traditional lumbar-focused treatments and the frequent absence of clear correlative findings on lumbar spine MRI [13].

Historically, “piriformis syndrome” was the default diagnosis for such cases, based on the early work of Yeoman and later popularized by Robinson [14,15]. This term, however, implies a muscular etiology that does not account for the diverse array of potential entrapment points now known to exist. The evolving understanding of gluteal anatomy, fueled by advancements in endoscopic surgery and high-resolution cross-sectional imaging, has revealed a more complex and diverse pathophysiology [16]. The deep gluteal space is a three-dimensional compartment with defined boundaries. Superiorly, it is bounded by the inferior margin of the sciatic notch; anteriorly by the hip joint capsule and the ischium; laterally by the linea aspera and the greater trochanter; medially by the sacrotuberous ligament and the falciform fascia; and inferiorly by the proximal origin of the hamstring muscles [12]. Its contents are neurovascular and muscular. The key neurovascular structures include the sciatic nerve, the posterior femoral cutaneous nerve, the pudendal nerve and internal pudendal vessels, and the superior and inferior gluteal nerves and vessels [17]. These structures are intimately related to and course through the short external rotator muscles, which form the “floor” of the space and include the piriformis, superior and inferior gemellus, obturator internus, and quadratus femoris muscles [18].

While the role of muscular hypertrophy, fibrovascular bands, and ischiofemoral impingement in DGS has been described in the literature [19,20], the potential for ligamentous pathology to act as a primary compressive agent has been largely overlooked and remains a significant gap in the current clinical understanding. The sacrospinous ligament (SSL), a robust, fan-shaped fibrous structure extending from the lateral margins of the sacrum and coccyx to the ischial spine, is a key stabilizer of the sacroiliac joint and the posterior pelvic floor [21]. It forms the inferior boundary of the greater sciatic foramen. The sciatic nerve’s relationship to the SSL is intimate and constant; the nerve typically exits the pelvis inferior to the piriformis muscle and courses inferolaterally, immediately adjacent to the lateral border of the SSL, with an average distance of approximately 1.4 cm from the ischial spine [22,23]. This fixed anatomical relationship makes the nerve particularly vulnerable to any pathological process that alters the structure or volume of the ligament. Pathological changes in the SSL, such as calcification (the deposition of calcium salts within the ligament substance) or ossification (the formation of true bone within the ligament), could theoretically alter its biomechanical properties, increase its volume, and reduce elasticity, thereby physically encroaching upon the adjacent sciatic nerve [24,25]. This compression can be dynamic, exacerbated during hip movements that add tension to the ligament, such as flexion, adduction, and internal rotation.

The diagnostic workup for DGS has been revolutionized by high-resolution musculoskeletal ultrasound (MSK-US) and magnetic resonance neurography (MRN) [26,27]. MRN excels at visualizing intraneural edema, denervation changes in muscles, and ruling out mass lesions. However, MSK-US offers several distinct advantages: it is dynamic, allowing for real-time assessment of nerve mobility against adjacent structures during provocative maneuvers; it is cost-effective and readily accessible; it provides high-resolution imaging of superficial structures; and it enables precise, real-time guidance for diagnostic and therapeutic injections [28,29]. For these reasons, MSK-US is becoming the modality of choice for the initial evaluation and management of suspected peripheral nerve entrapments like DGS.

Treatment for DGS remains challenging and is often multimodal, requiring a tailored approach. First-line management typically includes activity modification, physical therapy focusing on core and hip stabilizer strengthening, and neuromodulating medications (e.g., gabapentinoids, tricyclic antidepressants) [30,31]. When conservative measures fail, image-guided interventional procedures are the mainstay. Corticosteroid injections around the sciatic nerve are commonly employed to reduce perineural inflammation, while hydrodissection—the injection of fluid to mechanically separate the nerve from adhesions—aims to restore neural gliding [32,33]. Prolotherapy, also known as regenerative injection therapy, involves the injection of irritant solutions (most commonly hyperosmolar dextrose) to stimulate a localized healing response in damaged connective tissues [34]. The proposed mechanism involves triggering a controlled inflammatory cascade that leads to fibroblast proliferation, collagen deposition, and ultimately, ligament and tendon strengthening and stabilization [35]. While prolotherapy has shown promise in treating ligamentous laxity and tendinopathies in other parts of the body, its application in DGS, particularly for a ligamentous cause, is novel and has not been previously described in the literature [36].

This case report presents the first comprehensive description of DGS caused primarily by SSL calcification, diagnosed definitively through a combination of clinical suspicion and dynamic MSK-US, and successfully treated with a series of ultrasound-guided prolotherapy injections. The primary aim of this report is to introduce a new pathological entity into the differential diagnosis of sciatica, thereby expanding the anatomical and pathological spectrum of DGS beyond the traditional muscle-centric model. Secondly, it aims to propose and describe a novel, targeted, and effective image-guided treatment strategy for this specific condition, highlighting the therapeutic potential of prolotherapy in entrapment neuropathies. Finally, this report seeks to underscore the indispensable role of MSK-US not only in diagnosis but also in guiding precise intervention, thereby improving patient outcomes in complex pain conditions.

## 2. Case Presentation

### 2.1. Patient Information and History

A 51-year-old peri-menopausal female (height 169 cm, weight 79 kg, body mass index [BMI] 27.6 kg/m^2^) presented to our musculoskeletal medicine outpatient clinic on 29 April 2025, with a primary and debilitating complaint of severe left-sided gluteal and posterior thigh pain of several months’ duration. The patient, who worked full-time in a sedentary occupation as an administrative assistant, described the onset of symptoms as insidious, beginning approximately four months prior to her visit as a mild, dull, and intermittent ache localized deep within the left inferior gluteal region. Initially, the pain was manageable and did not significantly interfere with her work or daily activities. However, the pain escalated dramatically in both severity and frequency over the month preceding her consultation.

She attributed this acute exacerbation to a specific incident that occurred precisely one month prior. While performing household chores, she was kneeling on the floor and repetitively bending forward at the waist and twisting to clean under furniture. During this prolonged and awkward posture, she experienced a sudden, sharp, and tearing sensation deep within her left buttock, which was immediately followed by the onset of intense, burning pain that radiated down the posterior aspect of her thigh. Prior to this event, her medical history was significant only for occasional, mild mechanical low back pain that was never formally investigated or treated and never involved radicular symptoms or neurological deficits. There was no history of significant trauma, previous hip or pelvic surgery, autoimmune disorders, or metabolic conditions that might predispose to calcification. Her family history was non-contributory.

The pain persisted and progressively worsened to the point where her daily life became difficult. She reported a prior diagnosis of lumbar disk disease. Over a course of approximately three months, she received multiple interventions including two lumbar transforaminal epidural steroid injections (with 40 mg triamcinolone each), daily oral NSAIDs (naproxen 500 mg twice daily), a series of 6 spinal manipulation sessions, and 4 sessions of extracorporeal shockwave therapy (ESWT) targeting the lumbar region. Despite this comprehensive regimen, she reported no significant improvement in her sciatic symptoms as measured by persistent VAS scores > 8/10. The pain characteristics were highly mechanical. Prolonged sitting—especially on firm surfaces—for more than ten minutes became intolerable, forcing her to constantly shift her weight or stand up during work meetings. Driving a car, particularly the act of pressing the clutch pedal, was exquisitely painful. Ascending stairs also markedly aggravated her symptoms. The only factors that provided modest respite were standing still, lying on her right side, or walking slowly on level ground for short durations. The unremitting nature of the pain significantly impaired her sleep quality, as she was unable to find a comfortable position, leading to frequent nighttime awakening. This subsequently impacted her concentration at work, reduced her professional productivity, and curtailed her participation in social and recreational activities, culminating in a significant overall reduction in her quality of life and prompting her seek specialized care.

### 2.2. Clinical Findings

On physical examination, the patient’s demeanor was visibly distressed and she moved with evident caution. Her gait was profoundly antalgic upon entering the consultation room. She demonstrated a classic compensated Trendelenburg gait on the left, characterized by a significantly shortened stance phase on the affected limb, a lateral trunk lean over the left hip during the single-leg support phase to reduce demand on the hip abductors, and a refusal to fully load the left lower extremity. She verbally rated her pain at its worst, which occurred during walking and sitting, as 10 out of 10 on the Visual Analog Scale (VAS), and her pain at rest as a constant 6/10.

The patient was exceptionally precise in localizing her pain. She described its epicenter as being deep within the left inferomedial quadrant of the buttock, just superior to the gluteal fold and medial to the ischial tuberosity. From this point, it radiated as a sharp, burning, electric shock-like pain along the posterior thigh, terminating at the popliteal fossa. Additionally, she reported intermittent, lancinating paresthesias—described as “pins and needles”—that shot down the lateral aspect of her calf and into the dorsum of her left foot, a description highly suggestive of irritation affecting the peroneal division of the sciatic nerve. The pain was constant but exhibited clear mechanical characteristics, worsening notably after sustained hip flexion (e.g., sitting for >10 min), during the heel-strike phase of gait, and with active resistance. She also reported approximately 30 to 45 min of significant morning stiffness in the entire left posterior thigh and buttock region, which would gradually “loosen up” with movement.

Inspection with the patient standing and prone revealed no obvious muscle atrophy, limb length discrepancy, pelvic obliquity, or gross postural abnormalities. Detailed deep palpation of the left gluteal region, performed with the patient in a relaxed prone position, elicited exquisite, focal tenderness directly over the area of the ischial spine and the proximal sacrotuberous ligament. Notably, the tenderness was most intense deep to the bulk of the gluteus maximus muscle, requiring firm, deep pressure to elicit, and its reproduction caused the patient to recoil and confirm it was her “exact pain.”

A comprehensive neurological examination of the left lower limb was performed and was entirely within normal limits. Motor strength was graded as 5/5 in all major myotomes: hip extension (gluteus maximus, S1 nerve root), knee flexion (hamstrings, L5–S1), knee extension (quadriceps, L3–L4), ankle dorsiflexion (tibialis anterior, L4), great toe extension (extensor hallucis longus, L5), and ankle plantarflexion (gastrocnemius/soleus, S1). Deep tendon reflexes, including the patellar (L3–L4) and Achilles (S1) reflexes, were normoactive and symmetric (2+ bilaterally). Sensory testing to light touch, pinprick, and temperature was intact throughout the L4, L5, and S1 dermatomes, with no evidence of hyperesthesia or allodynia.

Examination of the lumbar spine revealed a full but somewhat guarded active range of motion. Forward flexion and lateral bending to the left reproduced a dull ache in the left low back but did not elicit the radicular component of her pain. Sacroiliac joint (SIJ) provocation tests were performed to rule out a concomitant pain generator. The Posterior Distraction (Gapping) Test and the Thigh Thrust (Piedallu’s) Test both reproduced her familiar deep left buttock pain (familiar symptom reproduction), yielding positive results.

The core of the examination focused on specific tests for DGS. The Flexion, Adduction, Internal Rotation (FAIR) test, a passive dynamic maneuver performed in the seated position, instantly reproduced her exact radiating pain pattern down the posterior thigh. The Pace Sign, which assesses for pain and weakness with resisted active abduction and external rotation of the hip in a seated position, was also positive, demonstrating clear weakness and pain on the left. Similarly, the Seated Piriformis Stretch Test (maximal passive internal rotation and adduction of the flexed hip while seated at the edge of the exam table) provoked severe, sharp gluteal pain. The Straight Leg Raise (SLR) test was attempted but was deemed “not testable” due to severe pre-existing pain and protective muscle guarding with minimal elevation of the limb (less than 20 degrees from the exam table).

### 2.3. Diagnostic Assessment

The constellation of findings—particularly the classic radicular pain pattern in the presence of a completely normal neurological exam, the positive cluster of DGS-specific tests, and the clear mechanical aggravating factors—strongly pointed towards an extra-spinal, non-discogenic etiology for her sciatica. DGS was established as the primary working diagnosis. Although lumbar radiculopathy was considered in the differential, its likelihood was deemed significantly lower given the absence of any neurological correlates. The positive SIJ tests indicated a potential concurrent SIJ dysfunction or irritability, which is not uncommon as both conditions can share similar pain referral patterns and may exist concomitantly.

To investigate further and rule out bony pathology, plain radiographs of the pelvis (anteroposterior (AP) and oblique views) and lumbar spine (AP and lateral views) were obtained using a standard radiographic unit (Siemens Ysio Max, Erlangen, Germany). Images were acquired with the patient standing for pelvic views (AP: 70 kVp, 25 mAs; Oblique: 75 kVp, 30 mAs) and supine for lumbar views (AP: 75 kVp, 30 mAs; Lateral: 85 kVp, 40 mAs). The AP pelvis radiograph revealed a well-defined, ovoid calcific density adjacent to the left inferolateral margin of the sacrum, precisely at the expected anatomical location of the SSL’s sacral attachment (Figure 1A). No significant hip joint arthropathy was noted, though a small, unrelated calcification was incidentally noted at the right greater trochanter, likely representing a benign enthesopathy or trochanteric calcific tendinitis. The lateral lumbar view confirmed mild disk space narrowing at L5-S1 (Figure 1B), consistent with early degenerative changes, but no spondylolisthesis, dynamic instability, or other significant findings that could account for her severe symptoms.

To precisely characterize this calcification and its relationship to the sciatic nerve, a comprehensive Musculoskeletal Ultrasound (MSK-US) examination was performed utilizing a curvilinear probe (C3–5). The sciatic nerve was systematically traced in short-axis from the infrapiriformis space distally towards the subgluteal fold. At the level of the ischial spine, a conspicuous, hyperechoic focus with dense posterior acoustic shadowing was identified embedded within the fibers of the SSL. This focus measured approximately 10 mm in its long axis and 3 mm in the anterior–posterior dimension and was consistent with a macroscopic calcific deposit within the ligament substance. The sciatic nerve was observed as a hyperechoic structure and showed prominent morphologic alteration compared to the healthy side (Figure 2A). In transverse view, it appeared markedly swollen, with an increased anteroposterior diameter of approximately 12 mm (compared to 8 mm on the asymptomatic contralateral side). There was a pronounced loss of the normal heterogeneous fascicular echotexture, sonographic findings highly consistent with reactive neuritis, edema, and possible fibrosis secondary to chronic adjacent mechanical irritation (Figure 2A). Dynamic assessment during passive internal rotation and adduction of the hip (simulating the FAIR position) showed a clear reduction in the normal gliding motion of the sciatic nerve against the rigid, unyielding calcified ligament, providing a dynamic correlate to the entrapment.

### 2.4. Therapeutic Intervention

A definitive diagnosis of DGS secondary to SSL calcification with associated sciatic nerve irritation was established. The treatment options, including continued conservative management with physical therapy, an ultrasound-guided corticosteroid injection for the neuritis, hydrodissection to free the nerve, or prolotherapy to address the source (the ligament pathology), were discussed in detail with the patient. After a detailed discussion of the risks, benefits, and alternatives, the patient opted to proceed with a trial of ultrasound-guided prolotherapy, motivated by its potential to treat the underlying structural cause. Informed consent was obtained, emphasizing the novel nature of this approach for this specific condition.

The procedure was performed with the patient in the prone position. The left gluteal region was widely prepped with chlorhexidine solution and draped in a sterile manner. Using an Alpinion XC90 elite (ALPINION MEDICAL SYSTEMS Co., Ltd., Seoul, Republic of Korea) system with a curvilinear transducer, the calcification within the SSL and the adjacent swollen sciatic nerve were clearly identified and confirmed. Under real-time ultrasound guidance, a 23-gauge, 100 mm needle was advanced to the target using an in-plane lateral-to-medial approach, allowing for continuous visualization of the entire needle shaft and tip throughout its trajectory to maximize safety and accuracy. The needle tip was meticulously positioned within the substance of the ligament, directly at the site of the calcific deposit. After confirming negative aspiration for blood, using a fenestration technique, multiple small aliquots (approximately 0.2 mL each) of the injectate were injected into the ligament substance and at the ligament–bone interface (enthesis). The injectate was prepared by mixing 4 mL of 10% dextrose in water (D10W) with 1 mL of 2% lidocaine (without epinephrine), yielding a final concentration of approximately 0.4% lidocaine in a dextrose solution with an osmolality of ~555 mOsm/kg. The total volume of 5 mL was administered slowly and under low pressure over approximately 2–3 min. The needle tip was subtly redirected between each injection. The flow of the hyperechoic solution was observed in real time, ensuring adequate dispersion within the ligament fibers and the periligamentous area surrounding the calcification (Figure 3), with careful avoidance of direct intraneural injection. The patient reported an immediate sensation of pressure and a precise reproduction of her familiar radiating pain during the injection, which was interpreted as a confirmatory sign of accurate needle placement at the pathological site.

Post-procedure, the patient was advised to apply ice to the area for 15 min every two hours for the first 24 h to manage any potential post-injection inflammation and soreness. She was instructed to avoid strenuous activities, heavy lifting, or prolonged sitting for the next 48–72 h. However, she was encouraged to engage in gentle walking and light movement to prevent stiffness and promote circulation. A predictable transient flare-up in her baseline pain occurred over the subsequent 36 h, which was managed effectively with ice and acetaminophen. By the third day post-procedure, she reported a substantial reduction in her resting pain to a VAS of 4/10 and noted an improved ability to sit for slightly longer periods.

This injection protocol was repeated at two-week intervals for a total of three sessions. Each session followed an identical sterile protocol and injectate formula. Following the first prolotherapy session, a formal, graded physical therapy program was initiated with a certified physical therapist. The rehabilitation protocol was deliberately phased to align with the proliferative healing response induced by prolotherapy and to avoid aggravating the irritated nerve:**Initial Phase (Weeks 1–2, post-first injection):** The focus was on pain modulation and restoring baseline neuromuscular control of the lumbopelvic hip complex. Modalities such as hot packs and magnetic therapy were utilized. Therapeutic exercises were limited to isometric contractions, including gluteal sets (for abductors and adductors) and transverse abdominis activation, performed in supine and sitting positions.**Intermediate Phase (Weeks 3–4, post-second injection):** The program incorporated proprioceptive neuromuscular facilitation (PNF) patterns targeting the gluteus medius, gluteus maximus, and deep core stabilizers. Gentle, passive sciatic nerve gliding exercises (e.g., supine slider techniques) were introduced to improve neural mobility and reduce perineural adhesions.**Advanced Phase (Week 5 onwards, post-third injection):** The focus shifted to functional strengthening, dynamic stabilization, and a safe return to pre-morbid activities. This phase included closed-chain exercises (e.g., mini-squats, lunges), balance training (e.g., single-leg stance), and progressive eccentric loading. The patient was highly compliant with her home exercise program throughout all phases.

Follow-up radiographs obtained three months after the final prolotherapy session revealed that the previously noted calcific density at the left SSL had significantly diminished and was no longer clearly discernible (Figure 4A,B). This suggests that the prolotherapy protocol may have contributed to the resorption of the calcific deposit, contrary to our initial hypothesis that its mechanism was solely functional stabilization. The mild L5-S1 disk space narrowing remained unchanged.

### 2.5. Follow-Up and Outcomes

The patient was monitored closely over a three-month period following the final prolotherapy session. The formal outcome assessment, including VAS, physical examination, and provocative testing, was conducted at this 3-month follow-up visit. The clinical outcomes were assessed both subjectively and objectively. Her subjective pain scores demonstrated a marked improvement. Her resting VAS score decreased from 10/10 at the initial visit to 1/10 at the final follow-up. Her pain during previously provocative activities like prolonged sitting or walking for more than 30 min was reduced to a mere 2/10 and was described as a mild, dull ache rather than a sharp, disabling pain. The pronounced antalgic gait observed initially had completely resolved, and she demonstrated a normal, symmetrical, and efficient gait pattern at all speeds. Repeat physical examination revealed a full, pain-free range of motion of both the lumbar spine and the left hip joint. Most significantly, all previously positive provocative tests—the FAIR test, Pace sign, Seated Piriformis Stretch, and the SIJ tests—were now unequivocally negative, indicating a complete resolution of the sciatic nerve irritability and entrapment. The patient successfully returned to her full-time sedentary job without accommodations and reported no limitations in her activities of daily living, including driving and light exercise. On the Patient Global Impression of Change (PGIC) scale, a validated outcome measure, she rated her overall improvement as “90% better.” She was discharged from physical therapy with a maintenance home exercise program and advised to return on a prn basis.

Objective functional measures also improved significantly. The patient’s sit-to-stand test time (5 repetitions) improved from 15 s at baseline to 8 s at follow-up. Additionally, she could tolerate sitting on a firm surface for 120 min at follow-up compared to less than 10 min initially.

## 3. Discussion

Deep gluteal syndrome is increasingly recognized as a challenging diagnostic entity in orthopaedic practice [37]. This case report describes the first documented instance of SSL calcification as a primary etiological factor in DGS. It further reports on the application of ultrasound-guided prolotherapy as a successful treatment modality for this specific condition. This finding significantly expands the anatomical and pathological spectrum of DGS, moving beyond the traditional muscle-centric model to include ligamentous pathology as a key player in the differential diagnosis of extra-spinal sciatica. The favorable outcome achieved through a targeted approach highlights the importance of a precise anatomical diagnosis in guiding therapy.

The anatomical basis for this entrapment is well-founded and provides a clear mechanistic explanation for the patient’s symptoms. The sciatic nerve’s close and consistent relationship to the SSL places it at inherent risk for compression from any space-occupying lesion or structural alteration of the ligament [22,23]. The nerve exits the greater sciatic foramen inferior to the piriformis muscle and courses inferolaterally, with its tibial and common peroneal divisions typically passing in close proximity to the lateral margin of the SSL before descending between the ischial tuberosity and the greater trochanter. The average distance between the nerve and the ischial spine, the lateral attachment of the SSL, is a mere 1.4 cm, leaving little room for error [21]. While calcification of the SSL causing sciatic entrapment has been reported in a handful of cases, often associated with pelvic imbalance or advanced degeneration [18,29], isolated calcification without full calcification has not been previously implicated in the literature as a cause of DGS. Calcification, the deposition of calcium hydroxyapatite crystals within the ligament substance, fundamentally alters its biomechanical properties. It reduces tissue compliance, increases volume and rigidity, and creates a fixed, non-compressible point that can mechanically irritate, compress, and restrict the normal gliding of the adjacent mobile nerve [38,39]. This compression is often dynamic, exacerbated during hip movements that tension the ligament and nerve simultaneously, such as flexion, adduction, and internal rotation (the FAIR position). This pathomechanism perfectly explains the patient’s positive provocative tests and the dynamic ultrasound findings of reduced nerve mobility.

The role of MSK-US was paramount and arguably diagnostic in this case. It allowed for not only the static identification of the calcification but also the dynamic assessment of nerve mobility relative to this pathological structure—a critical advantage over static imaging modalities like MRI or CT [28,29]. The sonographic findings of nerve swelling (loss of the normal fascicular architecture, diffuse hypoechogenicity, and thickening) provided objective, irrefutable evidence of neuritis and edema directly adjacent to the calcification. This spatial relationship, visualized in real-time, strengthens the causal link beyond mere correlation and helps exclude other potential causes [40]. Furthermore, MSK-US allowed us to rule out other common causes of DGS during the same examination, such as piriformis muscle hypertrophy, fibrovascular bands, or ischiofemoral narrowing. This case powerfully underscores MSK-US’s superiority as a first-line imaging modality for diagnosing dynamic entrapment neuropathies in the deep gluteal space, as it seamlessly integrates anatomical assessment with functional evaluation. It is important to note that while MRI and CT remain invaluable imaging modalities for assessing soft tissue structures, excluding mass lesions, and providing detailed anatomical characterization, MSK-US was prioritized in this case for its dynamic assessment, real-time symptom correlation, and interventional guidance. Future studies may benefit from a multimodal imaging strategy incorporating MRI or CT to further delineate ligamentous pathologies and corroborate ultrasound findings.

The success of prolotherapy in this case is mechanistically rational and suggests a paradigm shift from merely suppressing symptoms to addressing the presumed source of the problem: the weakened or pathologically altered ligament. Traditional approaches for DGS include corticosteroid injections, which aim to transiently suppress perineural inflammation but do nothing to alter the underlying compressive anatomy and carry risks of tissue atrophy [32]. Hydrodissection aims to mechanically separate the nerve from adhesions but may not be durable if the primary tethering point is a rigid calcification [33]. In contrast, prolotherapy targets the ligament itself. The injection of a hyperosmolar dextrose solution is hypothesized to stimulate a localized, controlled inflammatory cascade through osmotic shock, which leads to platelet degranulation and an influx of inflammatory mediators, growth factors, and fibroblasts, ultimately resulting in collagen deposition, and ultimately, ligament and tendon strengthening and stabilization [34,35]. This process results in ligament strengthening, thickening, and stabilization. In this case, follow-up radiographs suggested that the calcific density had significantly diminished and was no longer clearly discernible. This unexpected finding indicates that prolotherapy may not only stabilize the ligamentous complex—reducing abnormal micromotion and mechanical irritation of the sciatic nerve—but also facilitate a gradual bio-resorptive process of the calcific deposit. The precise mechanism remains uncertain and likely involves both functional stabilization and biological modulation of the local tissue environment. The added local anesthetic provided immediate symptomatic relief by blocking nociceptive signals, improving patient buy-in for the subsequent rehabilitation. The success of this approach suggests that the primary pain generator was not just the inflamed nerve (the effect) but the dysfunctional, calcified ligament (the cause). The concomitant physical therapy program was essential to address the secondary muscle inhibition, core weakness, and altered movement patterns that had developed as a consequence of chronic pain.

This case also illustrates the critical importance of maintaining a broad differential diagnosis for sciatica and the perils of premature diagnostic closure. The concomitant findings of L5–S1 disk narrowing on X-ray and SIJ tenderness on exam could have easily misdirected the treatment focus towards the spine or SIJ, leading to a series of potentially ineffective treatments. However, the specific cluster of positive DGS tests (FAIR, Pace, Seated Piriformis Stretch), the normal neurological examination, and the precise localization of tenderness guided the investigation away from the spine and towards the deep gluteal space. This highlights the absolute necessity of a thorough and knowledgeable physical examination that includes specific tests for DGS in any patient with sciatica-like symptoms, especially those refractories to lumbar-focused treatments or with incongruent imaging findings. It reinforces the principle that pain in the leg does not always originate from the back.

### 3.1. Limitations

The primary limitation of this study is its nature as a single case report. While the temporal relationship between the intervention and the dramatic resolution of symptoms is strongly suggestive of efficacy, it does not prove causation definitively. It is impossible to completely rule out the possibility of spontaneous remission or a significant placebo effect, though the chronicity and severity of the symptoms prior to intervention make this less likely. Furthermore, the patient received a multi-modal intervention (prolotherapy and physical therapy), making it difficult to isolate the exclusive contribution of each component to the final outcome. Furthermore, the concurrent physical therapy program represents a significant confounding variable. While the temporal association and targeted nature of the prolotherapy strongly suggest it was the primary effective intervention, the contribution of physical therapy to neuromuscular re-education, core stabilization, and functional recovery cannot be entirely disentangled in this case report. A study design sequentially introducing treatments would be needed to isolate their individual effects. Finally, while ultrasound showed clear improvement in nerve swelling and dynamic mobility at follow-up, the calcification itself remained morphologically unchanged, which supports the theory that prolotherapy works via functional stabilization and modulation of the tissue environment rather than structural dissolution of the calcific deposit. Long-term follow-up would be needed to see if the calcification resolves or changes over many years.

### 3.2. Future Directions

This report identifies a novel pathological entity; consequently, it opens several critical and actionable avenues for future research. To move from a singular observation to generalizable knowledge, the following studies are warranted:

First, a prospective diagnostic cohort study is essential to establish the prevalence and clinical significance of this finding. Such a study would systematically evaluate a large cohort of patients (e.g., *n* > 100) presenting with refractory sciatica and clinical features of DGS using standardized musculoskeletal ultrasound (MSK-US) protocols to assess the sacrospinous and other deep gluteal ligaments. The primary outcome would be the prevalence of ligamentous calcification or ossification. Secondary outcomes would include characterizing the associated sonographic features (e.g., nerve cross-sectional area, dynamic impairment of gliding) and correlating these findings with patient demographics and clinical examination features. This would determine if SSL pathology is a rare anomaly or a more common, under-recognized entity.

Second, once the patient population is better defined, a randomized controlled trial (RCT) is urgently needed to establish evidence-based treatment. The gold standard would be a double-blind, sham-controlled RCT comparing ultrasound-guided prolotherapy to a standard intervention such as a corticosteroid injection and to a sham procedure (e.g., superficial saline injection) for patients with DGS confirmed to be secondary to SSL calcification. Primary outcomes should include validated patient-reported measures of pain and function (e.g., VAS, ODI, HIP) at short-, medium-, and long-term follow-ups (e.g., 1, 3, 6, and 12 months). Objective measures, such as repeat ultrasound assessment of nerve swelling and dynamic mobility, should be included as secondary outcomes. This will provide Level I evidence regarding the efficacy of prolotherapy for this specific indication.

Third, basic science and histopathological research could explore the cellular and molecular changes induced by dextrose prolotherapy in calcified ligaments and adjacent neural tissue to elucidate its precise mechanism of action in this context. Finally, the development of standardized diagnostic criteria and systematic ultrasound protocols for evaluating the SSL and other ligaments in the deep gluteal space is a necessary prerequisite for both clinical implementation and future research. This would ensure diagnostic accuracy and consistency across providers and institutions, enabling larger multi-center studies.

## 4. Conclusions

This case identifies SSL calcification as a novel and previously underappreciated cause of DGS. It demonstrates that a ligamentous structure, often overlooked in the differential diagnosis of sciatica, can be a primary “pathological player” in sciatic nerve entrapment. This report expands the clinical and anatomical understanding of DGS, urging clinicians to look beyond muscles and disks. High-resolution musculoskeletal ultrasound is an indispensable tool for making this diagnosis, as it allows for dynamic assessment of the nerve-ligament interaction and provides precise guidance for intervention. In this case, ultrasound-guided prolotherapy targeting the pathological ligament was associated with a successful outcome, leading to functional recovery and resolution of symptoms. Physicians should therefore consider ligamentous entrapment in the differential diagnosis of refractory sciatica and utilize MSK-US to perform a comprehensive evaluation of the deep gluteal space. This case introduces a new diagnostic consideration and provides a potential new target for therapeutic intervention, which may improve outcomes for a subset of patients who have previously been misdiagnosed or failed traditional treatments.

## Figures and Tables

**Figure 1 life-15-01486-f001:**
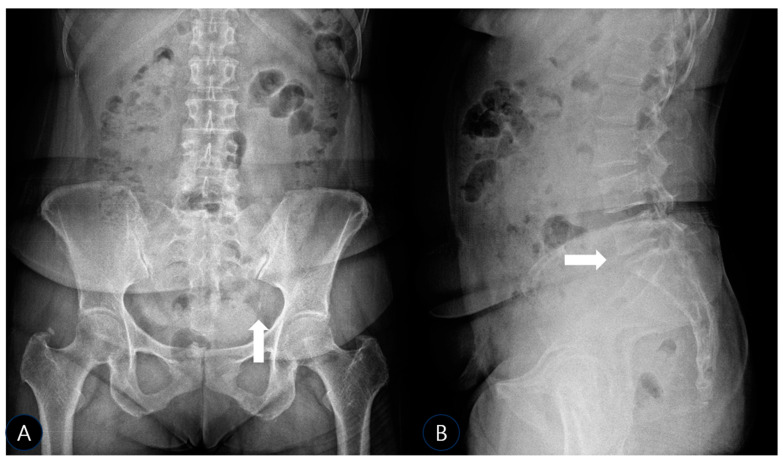
Lumbar AP, Lateral X-ray (**A**) Anteroposterior radiograph of the pelvis showing a well-defined calcific density at the left inferolateral sacrum (white arrow), corresponding to the anatomical location of the sacrospinous ligament. (**B**) Lateral lumbar spine radiograph demonstrating narrowing of the L5–S1 intervertebral disk space (white arrow), consistent with mild degenerative changes.

**Figure 2 life-15-01486-f002:**
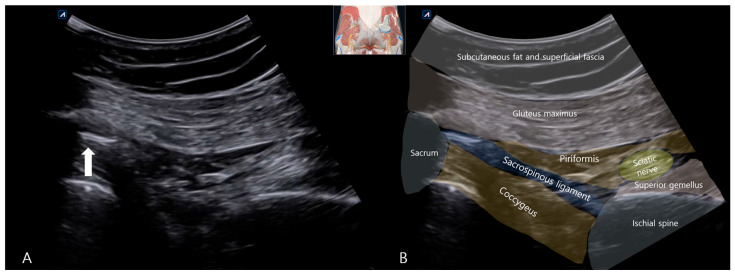
(**A**) Musculoskeletal ultrasound (MSK-US) image of the left sacrospinous ligament in long-axis view, demonstrating a hyperechoic calcific deposit (arrow) with dense posterior acoustic shadowing. The adjacent sciatic nerve appears swollen and hypoechoic compared to the contralateral side. (**B**) Schematic illustration of the anatomical relationship between the calcified ligament and the sciatic nerve for better localization. (The probe position is indicated in a schematic drawing on the ultrasound image).

**Figure 3 life-15-01486-f003:**
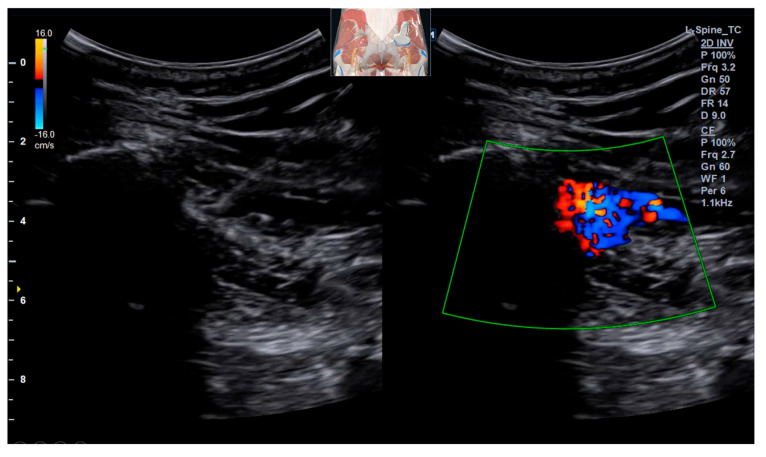
Ultrasound-guided prolotherapy procedure. Real-time live dual US imaging demonstrates the needle trajectory advancing into the sacrospinous ligament under an in-plane lateral-to-medial approach. The injectate (dextrose solution) is seen dispersing around the calcified ligament while carefully avoiding direct intraneural injection. (The probe position is indicated in a schematic drawing on the ultrasound image).

**Figure 4 life-15-01486-f004:**
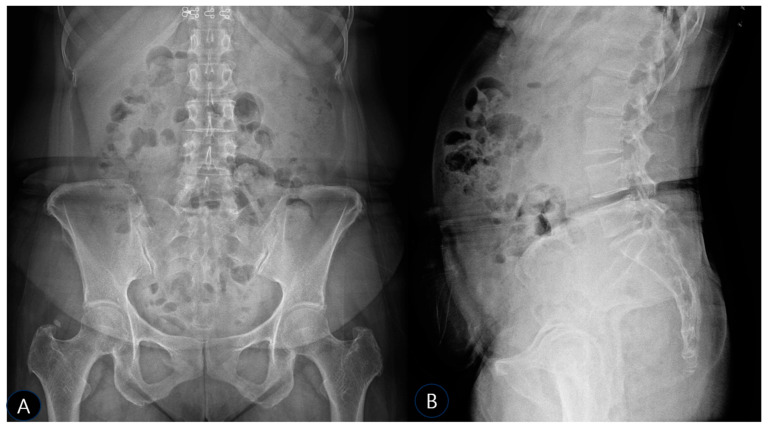
Lumbar AP, Lateral X-ray (**A**) In the AP image, it can be confirmed that the calcification present at the initial examination has disappeared. (**B**) In the lateral view, no changes in the disk space were observed, similar to those observed in the initial examination.

## Data Availability

The data presented in this study are available on request from the corresponding author (Y.Y., mgyyh00@gmail.com or K.H.S.L., driamkh@gmail.com). The data are not publicly available due to privacy and ethical restrictions.

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
