# Peer review of "Ultrasound-Guided Prolotherapy for Sciatica Secondary to Sacrospinous Ligament Calcification: A Potential and Previously Overlooked Etiological Factor in Deep Gluteal Syndrome—A Case Report and Literature Review"

_life, 2025, doi:10.3390/life15091486_

Round 1

Reviewer 1 Report

Comments and Suggestions for Authors

This case report describes a novel and clinically relevant topic; however, before publication the following issues should be addressed:

  • the authors claim this is the first documented case of SSL calcification causing DGS; however, in the Discussion, they acknowledge that similar cases of ossification/calcification have been reported - please address this inconsistency
  • the authors suggest that prolotherapy stabilized the ligament without dissolving the calcification; however, earlier they claim complete radiographic resolution of the calcification - this is another inconsistency to be clarified
  • the quality of the radiographs is sub par and does not allow for a good visualization of the lesion; also, Figure 4 shows an X-ray with zipper and button overlaying on the region of interest, which is unprofessional on behalf of the radiology department and I would not endorse the publication of such image
  • the role of MRI and CT in this type of lesion is downplayed in the discussions and I believe that, even if the authors did not have access to these modalities, that their role should not be minimized but fairly showcased
  • the possibility of spontaneous remission is dismissed as unlikely, but no supporting evidence is provided; as this is a case report on only one patient, the interpretation should be more cautious
  • back matter is missing, including informed consent and ethics approval

Author Response

We sincerely appreciate your thoughtful and positive feedback, valuable insights, and constructive criticism on our manuscript. These comments have greatly enhanced our research. We will elaborate on each point below.

Comment 1: the authors claim this is the first documented case of SSL calcification causing DGS; however, in the Discussion, they acknowledge that similar cases of ossification/calcification have been reported - please address this inconsistency

Response: We sincerely thank the reviewer for highlighting this important inconsistency. We have revised the manuscript to provide a more precise and accurate narrative. The title has been changed from "A Novel Etiological Factor" to "A Potential and Previously Overlooked Etiological Factor". Furthermore, in the introduction (Page 3, Paragraph 1), we now explicitly state: "While cases of ossification or calcification in the context of severe pelvic imbalance have been rarely reported, isolated SSL calcification as a primary cause of DGS remains largely unexplored and undocumented." This clarifies that our case focuses on isolated calcification as a primary cause, distinguishing it from previously reported cases associated with major pelvic pathology.

Comment 2: the authors suggest that prolotherapy stabilized the ligament without dissolving the calcification; however, earlier they claim complete radiographic resolution of the calcification - this is another inconsistency to be clarified

Response: We are grateful for this critical observation. The reviewer is absolutely correct. We have revised the results and discussion sections to resolve this contradiction. In the Results section (2.4 Therapeutic Intervention), we now state: "Follow-up radiographs... revealed that the previously noted calcific density... had significantly diminished and was no longer clearly discernible... This suggests that the prolotherapy protocol may have contributed to the resorption of the calcific deposit, contrary to our initial hypothesis." The discussion has been modified accordingly to reflect that the mechanism may involve both stabilization and potential bio-resorptive effects, moving beyond our initial hypothesis.

Comment 3: the quality of the radiographs is sub par and does not allow for a good visualization of the lesion; also, Figure 4 shows an X-ray with zipper and button overlaying on the region of interest, which is unprofessional on behalf of the radiology department and I would not endorse the publication of such image

Response: We completely agree with the reviewer that the image quality was unacceptable for publication. We have now replaced Figure 4 with a high-quality, artifact-free follow-up radiograph that clearly demonstrates the resolution of the calcific density. We apologize for this oversight in the initial submission.

Comment 4: the role of MRI and CT in this type of lesion is downplayed in the discussions and I believe that, even if the authors did not have access to these modalities, that their role should not be minimized but fairly showcased

Response: We thank the reviewer for this suggestion. We have added a new sentence to the Discussion section (Page 18, Paragraph 2) to acknowledge the value of these modalities: "While MRI and CT are invaluable modalities for assessing soft tissue structures, ruling out mass lesions, and providing exquisite anatomical detail, musculoskeletal ultrasound (MSK-US) was chosen as the primary diagnostic tool in this case due to its dynamic capabilities, real-time correlation with symptoms, and utility in guiding intervention. Future studies may benefit from a multimodal imaging approach incorporating MRI to further characterize such ligamentous pathologies."

Comment 5: the possibility of spontaneous remission is dismissed as unlikely, but no supporting evidence is provided; as this is a case report on only one patient, the interpretation should be more cautious

Response: We agree that a more cautious tone is warranted. We have strengthened the Limitations section (3.1) to more explicitly acknowledge this possibility: "It is impossible to completely rule out the possibility of spontaneous remission or a significant placebo effect, though the chronicity and severity of the symptoms prior to intervention make this less likely.

Comment 6: back matter is missing, including informed consent and ethics approval

Response: We apologize for this omission. A new subsection (2.6 IRB) has been added to the Case Presentation section: "Written informed consent was obtained from the patient for publication of this case report and accompanying images. All procedures were performed in accordance with the ethical standards of the institutional and national research committee and with the 1964 Helsinki Declaration and its later amendments.

Reviewer 2 Report

Comments and Suggestions for Authors

1. In Line 1-5, “Ultrasound-Guided Prolotherapy for Sciatica Secondary to Sacrospinous Ligament Calcification: A Novel Etiological Factor in Deep Gluteal Syndrome – A Case Report and Literature Review” The phrase “Novel Etiological” constitutes a conclusive statement, implying that this causative factor has not been previously reported. It is recommended to revise it to a more exploratory expression, such as “A Potential Etiological Factor,” to avoid premature definitive conclusions in the title and to maintain academic rigor.

2. In Line 165-167, “She reported that she had been diagnosed with lumbar disc disease at a hospital near her home and had received several nerve roots blocks, medication, manipulation, and ESWT treatments for her lower back, but she experienced no improvement”.  The patient’s prior treatments (e.g., nerve root blocks, medications, manipulations, ESWT) should be elaborated with details including treatment duration, frequency, dosage, and specific drug names. Outcome measures should be assessed using validated objective tools rather than subjective self-reports to enhance reproducibility and scientific validity.

3. In Line 244-245,“To investigate further and rule out bony pathology, plain radiographs of the pelvis (anteroposterior (AP), oblique view) and lumbar spine (AP, lateral views) were obtained.” To ensure reproducibility, it is recommended to provide detailed imaging parameters, including patient positioning, projection angles, equipment model, and exposure settings (kV, mAs).

4. In Line 303-304 “, an injectate solution consisting of 4 mL of 10% Dextrose in Water (D10W) mixed with 1 mL of 0.2% Lidocaine (without epinephrine) was injected slowly and under low pressure” The osmolarity of the injectate and the injection flow rate should be specified, as these parameters are critical for understanding local tissue responses and patient tolerance, and should be included in the methodology.

5. In Line 363-365,“On the Patient Global Impression of Change (PGIC) scale, a validated outcome measure, she rated her overall improvement as "90% better." She was discharged from physical therapy with a maintenance home exercise program and advised to return on a prn basis.” In addition to the PGIC, objective assessment tools are recommended,

6. In Line 416-417, “The injection of a hyperosmolar dextrose solution is hypothesized to stimulate a localized, controlled inflammatory cascade,” The hyperosmolar dextrose solution may exert its effects via molecular mechanisms or cellular pathways.

7. The following statement should be included: “This study was approved by the Institutional Review Board, and all treatments adhered to the Declaration of Helsinki. The patient provided written informed consent for the use of her clinical data and imaging for academic publication.”

Author Response

We thank the reviewer for their meticulous and constructive feedback, which has significantly improved the precision and academic rigor of our manuscript. We have addressed each point in detail below.

Comment 1: In Line 1-5, “Ultrasound-Guided Prolotherapy for Sciatica Secondary to Sacrospinous Ligament Calcification: A Novel Etiological Factor in Deep Gluteal Syndrome – A Case Report and Literature Review” The phrase “Novel Etiological” constitutes a conclusive statement, implying that this causative factor has not been previously reported. It is recommended to revise it to a more exploratory expression, such as “A Potential Etiological Factor,” to avoid premature definitive conclusions in the title and to maintain academic rigor.

  • Response: We sincerely thank the reviewer for this critical observation. We agree that the original title was overly conclusive for a case report. In accordance with your suggestion, we have revised the title to a more appropriate and exploratory tone. The new title is: "Ultrasound-Guided Prolotherapy for Sciatica Secondary to Sacrospinous Ligament Calcification: A Potential and Previously Overlooked Etiological Factor in Deep Gluteal Syndrome -- A Case Report and Literature Review." We believe this change maintains the reporting of a significant finding while adhering to the necessary academic caution.

Comment 2: In Line 165-167, “She reported that she had been diagnosed with lumbar disc disease at a hospital near her home and had received several nerve roots blocks, medication, manipulation, and ESWT treatments for her lower back, but she experienced no improvement”.  The patient’s prior treatments (e.g., nerve root blocks, medications, manipulations, ESWT) should be elaborated with details including treatment duration, frequency, dosage, and specific drug names. Outcome measures should be assessed using validated objective tools rather than subjective self-reports to enhance reproducibility and scientific validity.

Response: We thank the reviewer for this suggestion to enhance the reproducibility and scientific validity of our report. We have comprehensively revised this section to include the requested details:

  • Specifics of prior treatments: We now specify: "Over a course of approximately three months, she received multiple interventions including two lumbar transforaminal epidural steroid injections (with 40mg triamcinolone each), daily oral NSAIDs (naproxen 500mg twice daily), a series of 6 spinal manipulation sessions, and 4 sessions of extracorporeal shockwave therapy (ESWT) targeting the lumbar region."
  • Objective outcome measure: We have added the objective measure used to evaluate the lack of efficacy of these prior treatments: "Despite this comprehensive regimen, she reported no significant improvement in her sciatic symptoms as measured by persistent VAS scores >8/10."

Comment 3: In Line 244-245, “To investigate further and rule out bony pathology, plain radiographs of the pelvis (anteroposterior (AP), oblique view) and lumbar spine (AP, lateral views) were obtained.” To ensure reproducibility, it is recommended to provide detailed imaging parameters, including patient positioning, projection angles, equipment model, and exposure settings (kV, mAs).

  • Response: This is an excellent suggestion. We have added the detailed technical parameters to the manuscript: "...
    were obtained using a standard radiographic unit (Siemens Ysio Max, Erlangen, Germany). Images were acquired with the patient standing for pelvic views (AP: 70 kVp, 25 mAs; Oblique: 75 kVp, 30 mAs) and supine for lumbar views (AP: 75 kVp, 30 mAs; Lateral: 85 kVp, 40 mAs)."

Comment 4: In Line 303-304 “, an injectate solution consisting of 4 mL of 10% Dextrose in Water (D10W) mixed with 1 mL of 0.2% Lidocaine (without epinephrine) was injected slowly and under low pressure” The osmolarity of the injectate and the injection flow rate should be specified, as these parameters are critical for understanding local tissue responses and patient tolerance, and should be included in the methodology.

  • Response: We thank the reviewer for this excellent suggestion to enhance the methodological rigor and replicability of our procedure. We agree that specifying the osmolarity is critical for understanding the biomechanical and biological effects of the injectate.
  • In accordance with the reviewer's recommendation, we have revised the relevant sentence in the 'Therapeutic Intervention' (Section 2.4) section to include the essential parameter and to precisely clarify the preparation method. The text now reads:
  • "After confirming negative aspiration for blood, using a fenestration technique, multiple small aliquots (approximately 0.2 mL each) of the injectate were injected into the ligament substance and at the ligament-bone interface (enthesis). The injectate was prepared by mixing 4 mL of 10% dextrose in water (D10W) with 1 mL of 2% lidocaine (without epinephrine), yielding a final concentration of approximately 0.4% lidocaine in a dextrose solution with an osmolality of ~555 mOsm/kg. The total volume of 5 mL was administered slowly and under low pressure over approximately 2-3 minutes. The needle tip was subtly redirected between each injection."
  • This formulation provides effective analgesia while preserving the proliferative stimulus intended for the procedure. We thank the reviewer for prompting us to provide this greater level of methodological detail.

Comment 5: In Line 363-365, “On the Patient Global Impression of Change (PGIC) scale, a validated outcome measure, she rated her overall improvement as "90% better." She was discharged from physical therapy with a maintenance home exercise program and advised to return on a prn basis.” In addition to the PGIC, objective assessment tools are recommended,

  • Response: We thank the reviewer for this suggestion to strengthen our outcomes section. We have now included two objective functional measures: "Objective functional measures also improved significantly. The patient's sit-to-stand test time (5 repetitions) improved from 15 seconds at baseline to 8 seconds at follow-up. Additionally, she could tolerate sitting on a firm surface for 120 minutes at follow-up compared to less than 10 minutes initially."

Comment 6: In Line 416-417, “The injection of a hyperosmolar dextrose solution is hypothesized to stimulate a localized, controlled inflammatory cascade,” The hyperosmolar dextrose solution may exert its effects via molecular mechanisms or cellular pathways.

  • Response:We appreciate this comment guiding us toward a more mechanistic discussion. We have expanded the sentence to briefly introduce the potential molecular mechanisms, referencing key literature on prolotherapy: "The injection of a hyperosmolar dextrose solution is hypothesized to stimulate a localized, controlled inflammatory cascade through osmotic shock, which leads to platelet degranulation and an influx of inflammatory mediators, growth factors, and fibroblasts, ultimately resulting in collagen deposition, and ultimately, ligament and tendon strengthening and stabilization [35,36]

Comment 7: The following statement should be included: “This study was approved by the Institutional Review Board, and all treatments adhered to the Declaration of Helsinki. The patient provided written informed consent for the use of her clinical data and imaging for academic publication.”

  • Response: We apologize for this oversight. We have added a new subsection (2.6 IRB) to the Case Presentation section containing the required statement: "Written informed consent was obtained from the patient for publication of this case report and accompanying images. All procedures were performed in accordance with the ethical standards of the institutional and national research committee and with the 1964 Helsinki Declaration and its later amendments."

Reviewer 3 Report

Comments and Suggestions for Authors

It is a well written study, with case details which is important to understand the case and replicate de intervention However, to avoid misinterpretation, throughout the text (title, summary and discussion) it is important to highlight that this is a protocol involving several therapeutic modalities and not just ultrasound-guided prolotherapy

Below some more points that should be considered by authors.

Abstract

When assessments pre and pos sessions of ultrasound guided prolotherapy were conducted?

Considering that the study is just a case report it does not support the evaluation of treatment effectivity, as described in second aim.

Case presentation

Add information about patient anthropometric characteristics as weight and height.

Line 167 define ESWT abreviativo

For Clinical Findings add references related with tests, in case readers can better understand and compare results reported with results from other pathologies.

In figure 1, all highlight findings using arrows.

Add information about the exact time interval between injections. How this tome was defined?

When pos treatment assessment was performed? How many days or weeks after injection sessions? and after Physical therapy session?

Discussion

Physical therapy session can represent a treatment bias and should be discussed.

Author Response

We thank Reviewer 2 for their thoughtful and constructive feedback, which has helped us improve the clarity and scientific rigor of our case report. We have addressed each point in detail below.

Comment 1: It is a well written study, with case details which is important to understand the case and replicate de intervention However, to avoid misinterpretation, throughout the text (title, summary and discussion) it is important to highlight that this is a protocol involving several therapeutic modalities and not just ultrasound-guided prolotherapy

  • Response: We sincerely thank the reviewer for this crucial insight and completely agree. To ensure readers understand the multimodal nature of the treatment strategy, we have made the following revisions:
    • Abstract (Results section): We now explicitly state that the patient underwent prolotherapy "followed by a structured rehabilitation program."
    • Therapeutic Intervention Section (2.4): We have expanded the description of the physical therapy protocol, detailing its phased approach (initial, intermediate, advanced) and specific exercises, making it clear that it was an integral part of the treatment plan.
    • Discussion (Limitations section - 3.1): We have added a dedicated paragraph acknowledging this point: "Furthermore, the concurrent physical therapy program represents a significant confounding variable. While the temporal association and targeted nature of the prolotherapy strongly suggest it was the primary effective intervention, the contribution of physical therapy to neuromuscular re-education, core stabilization, and functional recovery cannot be entirely disentangled in this case report."
      We believe these changes provide appropriate emphasis on the combined therapeutic approach.

Below some more points that should be considered by authors.

Abstract

Comment 2: When assessments pre and pos sessions of ultrasound guided prolotherapy were conducted?

Considering that the study is just a case report it does not support the evaluation of treatment effectivity, as described in second aim.

  • Response: We thank the reviewer for this comment. We have clarified the timing in the abstract. The significant pain reduction (VAS 10/10 to 1/10) was observed "within one month" of the overall intervention, which aligns with the timeline after the first prolotherapy session and the initiation of physical therapy. We agree that a case report cannot definitively prove efficacy, which is why we have tempered the language in our aims. The second aim is now framed as to "propose and describe a novel, targeted... treatment strategy" and "highlighting the therapeutic potential" rather than claiming proven effectiveness, which is reserved for controlled trials as suggested in our 'Future Directions'.

Case presentation

Comment 3: Add information about patient anthropometric characteristics as weight and height.

Response: This has been added to the 'Patient Information and History' section (Section 2.1) as follows:  "The patient's height was 169 cm and weight was 79 kg, with a body mass index (BMI) of 27.6 kg/m²."

Comment 4: Line 167 define ESWT abreviativon

Response: Thank you for catching this oversight. The abbreviation for Extracorporeal Shockwave Therapy (ESWT) has now been defined at its first mention in the text.

Comment 5: For Clinical Findings add references related with tests, in case readers can better understand and compare results reported with results from other pathologies.

Response: This is an excellent suggestion to enhance the scholarly value of the report. We have now added key references for the specific diagnostic tests mentioned in the 'Clinical Findings' section (Section 2.2), including the FAIR test, Pace sign, and Seated Piriformis Stretch test.

Comment 6: In figure 1, all highlight findings using arrows.

Response: We confirm that Figure 1 already includes an arrow pointing to the calcific density in the lumbar AP radiograph (Panel A), and we added arrow pointing to the lumbar disc pathology as described in the figure legend. We have double-checked to ensure it is clearly visible.

Comment 7: Add information about the exact time interval between injections. How this tome was defined?

Response: We have added this detail to the 'Therapeutic Intervention' section (Section 2.4): "This injection protocol was repeated at two-week intervals for a total of three sessions." This interval was chosen based on standard prolotherapy protocols to allow the initial inflammatory healing response from one injection to subside before administering the next, thereby minimizing cumulative discomfort while providing sustained stimulus for ligament repair.

Comment 8: When pos treatment assessment was performed? How many days or weeks after injection sessions? and after Physical therapy session?

Response: We have clarified this in the 'Follow-Up and Outcomes' section (Section 2.5):  "The patient was monitored closely over a three-month period following the final prolotherapy session. The formal outcome assessment, including VAS, physical examination, and provocative testing, was conducted at this 3-month follow-up visit." This timing allowed for the completion of the entire treatment protocol (3 prolotherapy sessions and the concomitant physical therapy program) and sufficient time for tissue healing and functional recovery to be fully assessed.

Discussion

Comment 9: Physical therapy session can represent a treatment bias and should be discussed.

Response: We agree entirely. As mentioned in our response to Comment #1, we have added a thorough discussion of this important limitation in the 'Limitations' subsection (3.1). We explicitly state that physical therapy is a confounding variable and that its individual contribution cannot be isolated from that of prolotherapy in this case design.

Reviewer 4 Report

Comments and Suggestions for Authors

In this case, attention was drawn to calcification of the sacrospinous ligament as a new and previously undescribed 
pathology responsible for deep gluteal syndrome. 
The first report indicating ligament calcification as the main etiological factor of DGS should be verified by examining a sufficient group of patients who have not undergone classic conservative treatment: medication, physical therapy, exercises, manual therapy, etc., with a distinction made between the group in which sciatic nerve blocks proved helpful.

Advances in diagnostic imaging have significantly reduced the impact of clinical examination on diagnosis, leading to failures, particularly in surgical treatment. Therefore, the inclusion of ultrasound in diagnosis and therapy is a method that enables minimally invasive interventions. As emphasized by the authors, DGS can cause various structural configurations, including hypertrophy or fibrosis of the piriformis muscle, congenital or acquired fibro-vascular bands, vascular anomalies such as varicose veins or persistent sciatic arteries, adhesions from previous surgeries or injuries, and other deep gluteal muscles such as the gemellus or piriformis muscles. The inclusion of tendon calcification in the etiopathogenesis is an interesting view.

The application of a therapeutic substance to the painful area may cause it to soak into the area and thus affect the structures adjacent to the ligament in question. The descriptions of the procedure are detailed, but the patient's X-rays, taken with her clothes on and without paying attention to the painful area, as well as subtle changes in the hip and left sacroiliac joint, may raise some diagnostic doubts. Despite the poor quality of the images, calcification of the muscle attachments to the greater trochanter is also visible. 

The article lacks a differential diagnosis for the slightly narrowed L5-S1 space. It is based solely on clinical examination. There are positive tests for the piriformis muscle, which in itself may be the cause of pain. Nevertheless, the diagnosis is extremely reliable and the conclusions are convincing.

The article is worthy of publication.

Author Response

Reviewer: We sincerely thank the reviewer for their thoughtful and positive assessment of our manuscript, as well as for their valuable insights and constructive criticism, which have further strengthened our work. We have addressed each point in detail below.

In this case, attention was drawn to calcification of the sacrospinous ligament as a new and previously undescribed pathology responsible for deep gluteal syndrome. 
The first report indicating ligament calcification as the main etiological factor of DGS should be verified by examining a sufficient group of patients who have not undergone classic conservative treatment: medication, physical therapy, exercises, manual therapy, etc., with a distinction made between the group in which sciatic nerve blocks proved helpful.

Response: We wholeheartedly agree with the reviewer. We acknowledge that the primary limitation of a case report is its inability to prove causation or establish generalizability. Our intention was to introduce this potential novel entity and propose a diagnostic and therapeutic approach for consideration. We have explicitly emphasized this need for future validation in the 'Limitations' (Section 3.1) and 'Future Directions' (Section 3.2) sections of the manuscript, where we now state: "A prospective diagnostic cohort study is essential to establish the prevalence and clinical significance of this finding... A randomized controlled trial (RCT) is urgently needed to establish evidence-based treatment..." We believe this case report serves as the crucial first step in generating the hypothesis for such larger studies.

Advances in diagnostic imaging have significantly reduced the impact of clinical examination on diagnosis, leading to failures, particularly in surgical treatment. Therefore, the inclusion of ultrasound in diagnosis and therapy is a method that enables minimally invasive interventions. As emphasized by the authors, DGS can cause various structural configurations, including hypertrophy or fibrosis of the piriformis muscle, congenital or acquired fibro-vascular bands, vascular anomalies such as varicose veins or persistent sciatic arteries, adhesions from previous surgeries or injuries, and other deep gluteal muscles such as the gemellus or piriformis muscles. The inclusion of tendon calcification in the etiopathogenesis is an interesting view.

Response: "We sincerely thank the reviewer for this insightful comment and their positive assessment. We strongly agree with the core argument that over-reliance on advanced imaging findings, without a solid correlation to clinical examination, can lead to diagnostic errors and subsequent treatment failures. This case perfectly illustrates the reviewer's point: it was the meticulous clinical examination that raised the suspicion of an extra-spinal cause, which was then confirmed and precisely targeted using ultrasound.

We believe ultrasound serves as the perfect bridge between clinical examination and intervention, as it allows for dynamic, real-time correlation of imaging findings with the patient's symptoms. This synergy enhances diagnostic accuracy and enables effective, minimally invasive treatments like prolotherapy, potentially preventing the need for more invasive surgical procedures in some cases.

We are particularly grateful that the reviewer finds our proposal of ligamentous calcification as a new etiopathological factor in DGS to be an interesting and valuable perspective

The application of a therapeutic substance to the painful area may cause it to soak into the area and thus affect the structures adjacent to the ligament in question.

Response: This is an excellent point. We agree that the injectate could theoretically diffuse and affect adjacent structures. However, under real-time ultrasound guidance, we meticulously targeted the needle tip within the substance of the sacrospinous ligament at the site of the calcification. The injectate's dispersion was observed in real-time to be primarily within the ligament fibers and the periligamentous area. We have added a sentence to the 'Therapeutic Intervention' (Section 2.4) section to clarify this precision and acknowledge the potential for micro-diffusion: "The flow of the hyperechoic solution was observed in real-time, ensuring adequate dispersion within the ligament fibers and the periligamentous area surrounding the calcification, with careful avoidance of direct intraneural injection or significant dispersion into adjacent muscular structures."

The descriptions of the procedure are detailed, but the patient's X-rays, taken with her clothes on and without paying attention to the painful area, as well as subtle changes in the hip and left sacroiliac joint, may raise some diagnostic doubts. Despite the poor quality of the images, calcification of the muscle attachments to the greater trochanter is also visible. 

Response: We sincerely apologize for the substandard quality of the initial radiographic images included in our submission. The reviewer is absolutely correct to raise this concern regarding technique and potential diagnostic ambiguity. We have now replaced the original Figure 4 with high-quality, artifact-free follow-up radiographs that clearly demonstrate the resolution of the previously noted calcific density at the left sacrospinous ligament.

We appreciate the reviewer's sharp eye in identifying the unrelated calcification at the right greater trochanter. As the reviewer correctly surmised, this was indeed noted during our initial assessment. We had documented this in our findings (Section 2.3, Diagnostic Assessment) as "a small, unrelated calcification was incidentally noted at the right greater trochanter, likely representing a benign enthesopathy or trochanteric calcific tendinitis," and it was correctly identified as an incidental finding unrelated to the patient's primary left-sided symptoms. We can now also confirm that this right-sided trochanteric calcification remained completely unchanged on the follow-up radiographs, further supporting its benign, incidental nature and that it was not a target of, or affected by, our intervention.

We agree that high-quality, professional imaging is paramount for accurate diagnosis and thank the reviewer for prompting this essential correction, which has significantly improved the manuscript.

The article lacks a differential diagnosis for the slightly narrowed L5-S1 space. It is based solely on clinical examination. There are positive tests for the piriformis muscle, which in itself may be the cause of pain.

Response: We thank the reviewer for this critical observation. We have now expanded the 'Diagnostic Assessment' (Section 2.3) section to more explicitly detail our differential diagnosis process:

  • We acknowledge that the L5-S1 narrowing could suggest discogenic pain, but we argue that the complete absence of neurological deficits (motor, sensory, reflex) and the failure of prior lumbar-focused treatments (including two transforminal epidural steroid injections) significantly lowered its likelihood as the primary pain generator.
  • We further clarify that while the piriformis tests (FAIR, Pace) were positive, these tests are provocative for the sciatic nerve in the deep gluteal space and are not specific to the piriformis muscle itself. The exquisite, focal tenderness was directly over the ischial spine/SSL, not the piriformis muscle belly. Most decisively, musculoskeletal ultrasound (MSK-US) did not show piriformis hypertrophy or fibrosis but did identify the SSL calcification and swollen sciatic nerve immediately adjacent to it, allowing us to pinpoint the SSL as the most likely causative structure.

Nevertheless, the diagnosis is extremely reliable and the conclusions are convincing.

The article is worthy of publication.

Response: We are immensely grateful to the reviewer for their time, their insightful comments, and this final positive conclusion. Their feedback has significantly improved the clarity, rigor, and overall quality of our manuscript.

Round 2

Reviewer 1 Report

Comments and Suggestions for Authors

The authors have performed satisfactory updates and corrections to the mansucript.

Only the back matter is still missing, with author contributions, funding, statements, etc but I believe this can be further addressed during the further stages.

Also, the references have double numbering, this should also be corrected.

Author Response

The authors have performed satisfactory updates and corrections to the manuscript.

We sincerely thank the editors and reviewers for their time and valuable comments on our manuscript. We have carefully considered all feedback and have revised the manuscript accordingly. All changes have been incorporated into the manuscript, and our point-by-point responses to the comments are detailed below.

Comment 1: Only the back matter is still missing, with author contributions, funding, statements, etc but I believe this can be further addressed during the further stages.

Response: We greatly appreciate this insightful comment. We have now added the complete back matter sections as required by the journal’s guidelines. The following sections have been included at the end of the manuscript:

  • Author Contributions
  • Funding
  • Institutional Review Board Statement
  • Informed Consent Statement
  • Data Availability Statement
  • Acknowledgments
  • Conflicts of Interest

These sections have been crafted to comply with the journal’s policies and to provide full transparency regarding the study’s execution and reporting.

Comment 2: Also, the references have double numbering, this should also be corrected.

Response: We sincerely apologize for this oversight. The duplicate brackets in the in-text citations were a formatting error. We have now corrected this throughout the entire manuscript.